# Hospital and Population-Based Evidence for COVID-19 Early Circulation in the East of France

**DOI:** 10.3390/ijerph17197175

**Published:** 2020-09-30

**Authors:** Laurent Gerbaud, Candy Guiguet-Auclair, Franck Breysse, Joséphine Odoul, Lemlih Ouchchane, Jonathan Peterschmitt, Camille Dezfouli-Desfer, Vincent Breton

**Affiliations:** 1Public Health Department, National Center for Scientific Research, University Hospital of Clermont-Ferrand, Clermont Auvergne University, SIGMA Clermont, Pascal Institute, 63000 Clermont-Ferrand, France; cauclair@chu-clermontferrand.fr (C.G.-A.); lemlih.ouchchane@uca.fr (L.O.); 2Emergency Department of Diaconat Fonderie Hospital, 68100 Mulhouse, France; franck.breysse@diaconat-mulhouse.fr (F.B.); desfercamille@yahoo.fr (C.D.-D.); 3Public Health Department, University Hospital of Clermont-Ferrand, 63000 Clermont-Ferrand, France; jodoul@chu-clermontferrand.fr; 4Sundgau Medical Center, 68210 Bernwiller, France; ulrichvonelsass@gmail.com; 5Laboratoire de Physique de Clermont, National Center for Scientific Research, National Institute of Nuclear and Particle Physics, Université Clermont Auvergne, F-63000 Clermont-Ferrand, France; Vincent.breton@clermont.in2p3.fr

**Keywords:** COVID-19, SARS-Cov-2, epidemic surveillance, emerging infectious disease, epidemic threshold

## Abstract

Background: Understanding SARS-CoV-2 dynamics and transmission is a serious issue. Its propagation needs to be modeled and controlled. The Alsace region in the East of France has been among the first French COVID-19 clusters in 2020. Methods: We confront evidence from three independent and retrospective sources: a population-based survey through internet, an analysis of the medical records from hospital emergency care services, and a review of medical biology laboratory data. We also check the role played in virus propagation by a large religious meeting that gathered over 2000 participants from all over France mid-February in Mulhouse. Results: Our results suggest that SARS-CoV-2 was circulating several weeks before the first officially recognized case in Alsace on 26 February 2020 and the sanitary alert on 3 March 2020. The religious gathering seems to have played a role for secondary dissemination of the epidemic in France, but not in creating the local outbreak. Conclusions: Our results illustrate how the integration of data coming from multiple sources could help trigger an early alarm in the context of an emerging disease. Good information data systems, able to produce earlier alerts, could have avoided a general lockdown in France.

## 1. Introduction

The coronavirus disease COVID-19 was labelled a pandemic by the World Health Organization (WHO) on 12 March 2020 [1]. At that time, there were more than 20,000 confirmed cases and almost 1000 deaths in Europe according to WHO statistics. France has been heavily affected by the SARS-CoV-2 epidemic [2]: the first three cases of COVID-19 in France were confirmed on 24 January 2020 [3]. The Oise department north of Paris was among the first COVID-19 clusters in France where SARS-CoV-2 was actively circulating weeks before the country lockdown on 17 March 2020 [4]. Alsace is a cultural and historical region on the border between France and Germany in the “Grand Est” administrative region. The Haut-Rhin department in the southern part of Alsace was also identified among the earliest COVID-19 clusters in France. The first official case of coronavirus in the Grand Est region, a traveler coming back from Lombardi (Italy), was identified on 26 February 2020 [5]. At the beginning of March, the number of cases increased rapidly, particularly among the participants of a religious gathering that took place from 17–21 February 2020 at Porte Ouverte Chrétienne (POC) Church in Mulhouse, a city of 110,000 inhabitants.

To understand the characteristics of the outbreak, the particular role played by the POC gathering in the virus transmission in Haut-Rhin and to see if the alert could have been triggered earlier, an online population-based survey was launched 22 April 2020 with the support of general physicians and emergency unit health professionals of the Haut-Rhin department [6]. Population-based surveys are frequently used in social sciences but also in epidemiology to study the prevalence of diseases in a specific human group or the impact of social patterns on the spread of infectious diseases, particularly on respiratory syndromes [7,8,9].

In this paper, we compare the results of this survey to those of a retrospective study led by an emergency unit located in Mulhouse in order to check their coherence. Indeed, emergency services have been on the front line of the pandemic, especially during the first weeks of March 2020, when they had to handle a massive increase of COVID-19 cases. Retrospective analysis of medical files in a French hospital North of Paris helped in identifying patients with COVID-19 symptoms as early as December 2019 [10].

Until March 2020, hospital biological laboratories were still performing PCR tests for influenza, while PCR tests for SARS-CoV-2 coronavirus were strictly restricted to persons coming back from regions at risk. In the context of emergency care, influenza PCR tests can be requested for patients displaying clinical symptoms very specific to influenza infection in order to confirm the diagnosis. If COVID-19 was already circulating in the Haut-Rhin population at that time, early COVID-19 cases displaying typical influenza symptoms should have been tested negative for influenza. As a consequence, the rate of negative influenza tests should also provide a useful piece of information to understand the beginning of the COVID-19 epidemic.

The objective of our study is to draw a coherent picture of the COVID-19 epidemic history in the Haut-Rhin department from January to April 2020 by confronting data coming from these three sources: an online population-based survey, an emergency care service retrospective study, and influenza PCR laboratory tests performed upon request of emergency care physicians.

## 2. Materials and Methods

### 2.1. Data Sources

#### 2.1.1. Online Population-Based Survey

A population-based survey using an anonymous questionnaire was conducted online to evaluate if people living in the Haut-Rhin department had symptoms commonly experienced in case of COVID-19 infection. In order to achieve a quick deployment to reduce memorization bias, a fast track procedure was followed to build the survey protocol and obtain the needed legal and ethical agreements.

After testing the questionnaire on four voluntary families in France and Switzerland, the study was approved by the legal and ethical agreements of Clermont Auvergne University Ethic Committee on 10 April 2020, and Clermont Auvergne University Personal Data Protection Management authorities and National Center for Scientific Research Personal Data Protection Management authorities both on 21 April 2020 (reference number IRB00011540-2020-37). Informed consent was obtained from all participants.

All families living in the Haut-Rhin department including POC meeting participants were invited to fill the online questionnaire. Thanks to the support of local media (local newspapers and radio) and to the POC meeting organizers, the survey was well advertised. There was no control nor deadline for participation.

The survey considered the family structure and the social relationships and networks of the household members. Respondents were then asked to document the occurrence among family members of the symptoms commonly observed in the case of COVID-19 infection [11]. The intensity, duration, and possible regrowth of each of the following clinical signs were documented: fever, cough, breathing difficulties, asthenia, loss of taste and/or smell, diarrhea, aches, Ear, Nose and Throat (ENT) symptoms, and neurological and dermatological symptoms. Respondents were also asked if they had been diagnosed with COVID-19, either through a PCR test or remotely, and if they had been tested for influenza. The survey structure allowed for a description of up to five possible COVID-19 cases in the same household. In addition, they were invited to indicate whether a household member had participated to the POC gathering in February 2020 in Mulhouse.

#### 2.1.2. Emergency Care Service

The second source of information is the emergency care service of the Diaconat-Fonderie Hospital in Mulhouse. This private, not-for-profit hospital is located in the center of Mulhouse and has a capacity of 200 beds. Its emergency care service was created in 2006: the staff includes 7 emergency physicians supported by 17 paramedics. In 2018, the service recorded 28,317 patients, to be compared to 68,552 patients at Mulhouse public hospital emergency service the same year.

The detection of COVID-19 patients became simpler in May 2020 thanks to the availability of PCR and serological tests combined with the clinical experience of healthcare professionals. This experience was used to reevaluate retrospectively all the medical records of patients who visited the emergency care service from 30 December 2019 to 18 May 2020 in order to identify potential COVID-19 cases.

For patients admitted before the local sanitary alert on 3 March 2020, the selection process involved several stages. In a first stage, medical records were preselected by the head nurse based on patient reason for emergency consultation. Records of patients presenting symptoms possibly related to a COVID-19 infection were kept, including fever, cough, dyspnea, headache, diarrhea, dizziness, or unexplained discomfort. Files were excluded if medical evaluation recorded at both entrance and exit was not compatible with COVID-19 and if PCR tests for influenza were positive. The preselected patient files were then analyzed by the emergency physicians and classified as probable COVID-19 cases if several of the following clinical conditions were fulfilled:The patient interview revealed previous contacts with COVID-19 symptomatic persons.The following clinical signs were present: significantly impaired general condition, fever, myalgia, arthralgia, dry cough, significant asthenia, dyspnea, desaturation, sibilant rales at auscultation, chest pain, headaches, unexplained discomfort, anosmia, diarrhea, or abdominal pain. Symptom duration was considered. Productive cough and rhinorrhea were considered symptoms excluding COVID-19.Chest computed tomography (CT) scans suggested COVID-19: the presence of Ground Glass Opacities [12]. Ambiguous CT-scan images were double-checked by radiologists. A clear pneumonia spot precluded COVID-19.Biological indicators were in favor of COVID-19: lymphopenia, a negative influenza PCR test, or thrombocytopenia.

For patients admitted after 3 March 2020, the head nurse reviewed all patient files and performed a first selection of potential COVID-19 cases based on, respectively, the reason for consulting, the nurse’s clinical assessment, and the results of medical examinations, with a particular focus on imaging and biological tests. The selected patients were further evaluated by an emergency physician who established a list of probable COVID-19 patients. This list was then revised by integrating data coming from the radiology service (scanner examinations) and the medical laboratory (biological tests). COVID-19 infection was retrospectively confirmed for medical records with chest images indicating typical CT features of COVID-19 pneumonia and/or a positive biological test for COVID-19. Patients were classified as confirmed cases only after 3 March 2020, when biological tests were available.

At the end of this process, 552 patients were retained as retrospectively diagnosed probable or confirmed cases, out of the 7184 patients admitted to the emergency care service from 29 December 2019 to 18 May 2020.

#### 2.1.3. Medical Biology Laboratory

The Diaconat-Fonderie medical biology laboratory is a multi-purpose biological platform open 24 h a day every day to provide continuous support to emergency care services. Its team handles over 600 daily samples coming from more than 20 healthcare centers in the Haut-Rhin department. The laboratory performed influenza PCR tests of the samples sent by the Diaconat-Fonderie emergency care services between 1 January and 15 March 2020 using the Abbott BinaxNOW Influenza A&B Card kit (Abbott, North Chicago, IL, USA) [13]. After the sanitary alert on the COVID-19 epidemic was raised and containment was established by the government, PCR tests for influenza were no longer performed.

### 2.2. Data Analysis

Statistical analyses were conducted using SAS software v9.4 (SAS Institute, Inc., Cary, NC, USA). Categorical variables were presented as frequencies and percentages, while continuous variables were presented as means and standard deviations (SD).

Overall epidemic evolution was compared in the survey data and in the retrospective analysis of emergency care service medical records by computing an epidemic threshold for both data sources.

A wide variety of statistical methods are available for defining an epidemic threshold for seasonal epidemics such as influenza [14,15,16]. In the case of emerging diseases, there is no criteria to determine how many infected individuals are needed to declare that an outbreak is occurring [17]. We chose to evaluate the epidemic threshold by applying a method similar to the one described for influenza-like syndromes [16]. We first computed a non-epidemic level using data collected before the epidemic started. The threshold was then defined as the upper 95% confidence limit of this non-epidemic level.

From the answers of Haut-Rhin households, we computed every day from 15 December 2019 to 16 May 2020 the number of new possible COVID-19 cases using the first day of illness onset. The non-epidemic level was defined as the highest value of the seven-day moving average of daily incidence over the period from 15 December 2019 to 15 January 2020. The epidemic threshold was then computed as the upper 95% confidence limit of the non-epidemic level.

The same approach was adopted for Diaconat-Fonderie medical records. We calculated every day from 30 December 2019 to 16 May 2020 the number of new possible COVID-19 cases using the entrance date in the emergency care service. The non-epidemic level was defined as the highest value of the seven-day moving average of daily COVID-19 cases over the period from 1 January to 15 January 2020.

In a second step, we checked for both datasets the stability in terms of consecutive days over or under the threshold: if the incidence was rapidly oscillating under and over the threshold, it meant that the threshold did not discriminate between outbreak and background noise. The signal was stable if each period under or over the threshold was larger than seven consecutive days. In order to complete the incidence curve analysis, we also computed the daily speed (Equation (1)) and acceleration (Equation (2)).
*s_i_* = Δ*m_i_*/Δ*t_i_*(1)
*a_i_* = Δ*s_i_*/Δ*t_i_*(2)
where *s* corresponds to the speed, *m to* the seven-day moving average of daily incidence, *t* to the time, and *i* to the day.

For the majority of cases documented in the population-based survey, biological proofs of SARS-CoV-2 infection were missing, and there could be confounding epidemic events, as documented for instance in [4]. To confirm the nature of the epidemic tracked and to measure its dynamic, we compared our results to the studies based on cohorts with proved biological infection. Therefore, using the intra-household data, we estimated the secondary attack rate, defined as the probability that an infection occurs among susceptible persons within a specific group [18]. It was approximated by the proportion of secondary cases induced in one household by the very first case of COVID-19 in the household, assuming that all the secondary cases were generated by a single primary case [19,20,21]. We also estimated the generation interval as the mean difference between the occurrence days of two successive cases in the same family [22,23]. Considering that children were more sensitive to other seasonal respiratory viruses than to COVID-19 [24,25], we compared the seven day moving average of daily incidence among children under 15 years old and among adults and children aged 15 years and older.

In order to further discriminate COVID-19 cases from other respiratory syndromes circulating in the population, we particularly focused on anosmia, identified as the symptom with the largest positive predictive value for SARS-CoV-2 infection [4,26,27,28]. The predictive value is as high as 80% in the presence of other respiratory syndromes [26]. If anosmia is quite specific to SARS-CoV-2 infection, it is nevertheless experienced by only 30% to 50% of the COVID-19-infected patients [4,26,27], sometimes less, as shown by a recent meta-analysis [29]. However, it was shown to be a relevant screening tool to help identify people with potential mild cases [28]. In the population survey, participants documenting possible COVID-19 cases were invited to evaluate this symptom intensity on a scale from 0 (no modification) to 3 (complete loss). Unfortunately, anosmia was not documented in the medical records of Diaconat-Fonderie emergency care service, as its relevance to COVID-19 diagnosis was not known at the beginning of the epidemic.

## 3. Results

### 3.1. Epidemic Evolution of Possible COVID-19 Cases

#### 3.1.1. Population-Based Survey

From 22 April to 4 June 2020, 1427 households participated to the population-based survey. The 201 households not living in the Haut-Rhin department were excluded from the present analysis, leading to a population of 1226 households, representing 3350 individuals. Among them, 883 households (72.0%), representing 2502 individuals, reported at least one possible case of COVID-19 (Table 1). A total number of 1516 individuals (including 26 of unknown age) experienced symptoms commonly attributed to COVID-19 infection, corresponding to 1301 adults and children aged 15 years and older and 189 children less than 15 years old. First possible cases were declared as early as November 2019.

Figure 1 shows the seven-day moving average of daily incidence based on illness onset day identified by the appearance of the first COVID-19 symptoms (blue line).

Based on the number of cases documented in the reference period (15 December 2019 to 15 January 2020), the epidemic threshold (dashed blue line) was estimated to 5 new cases per day. The curve shape indicates three waves in the epidemic development, which is confirmed by the speed and acceleration analysis (data not shown). From 30 December 2019 to 22 January 2020, a first wave lasted 24 days, with the same one-week increase (1–7 January) and decrease (17–24 January) periods. However, the highest incidence remained under the epidemic threshold. The second wave also lasted 24 days from 23 January to 17 February. It crossed the epidemic threshold on 26 January and remained over this threshold despite a decrease after 10 February. The third wave of the outbreak began on 17 February, with an incidence increase leading to a first maximum on 1 March, 3.6 times higher than the preceding wave. A partial and local lockdown started on 3 March and was extended to the whole country on 17 March [30]. The incidence began to decrease after 20 March, 17 days after the beginning of the local lockdown.

Among households documenting at least one possible COVID-19 case, 47.6% indicated that the onset of the first COVID-19 symptoms occurred before 1 March 2020. On 17 March, the first day of nationwide lockdown, 81.2% of these households had already reported a possible COVID-19 case. On the positive side, the lockdown was very efficient at stopping the epidemic as only 2.9% of the households reported their first case after 31 March, two weeks after entering the lockdown.

For the whole period, the secondary attack rate was 37.6%, and the mean generation interval, computed using intra-household data, was 4 days.

#### 3.1.2. Emergency Care Service

After analyzing the records of all the patients admitted into the Diaconat-Fonderie emergency care service from 30 December 2019 to 18 May 2020, 552 retrospectively diagnosed probable or confirmed COVID-19 cases were retained.

The seven-day moving average of daily incidence according to patient admittance day is documented on Figure 1 (yellow line). We estimated the epidemic threshold at 1.2 cases a day (dashed yellow line). The first case occurred on 6 January, and the threshold was crossed for the first time on 29 January. The first outbreak wave remained flat, although the incidence stayed over the threshold until 10 February, with a first weak maximum on 4 February. The incidence then remained under but close to the threshold, which was crossed once again on 28 February, showing an important outbreak, with a maximum on 27 March, which was 7.6 times higher than in February.

In order to check the possible role of emergency attendance, we also computed the fraction of the weekly attendance represented by retrospectively diagnosed probable or confirmed COVID-19 cases (Figure 2). The pattern observed in Figure 2 is close to the yellow line on Figure 1, showing a little increase at the end of January and a major increase after 24 February.

#### 3.1.3. Evolution of Influenza PCR Tests

In the period preceding the sanitary alert, the circulation of COVID-19 in the population of the Haut-Rhin department was ignored. Emergency care service visitors with specific influenza-like symptoms were given influenza PCR tests from January to mid-March 2020 to confirm diagnosis.

The emergency care services sent 59 samples to the Diaconat-Fonderie medical biology laboratory for influenza PCR testing from 1 January to 15 March 2020; 66% of the tests performed up to 28 January 2020 were positive.

In the population-based survey, 3.2% of the cases (48 out of 1516) were tested for influenza. Three-quarters of the tests were negative (n = 36). Out of the 12 persons tested positive for influenza, six also tested positive for COVID-19. Cases of coinfection by COVID-19 and influenza have been confirmed and documented [31]. As a consequence, we did not exclude the cases with influenza PCR positive tests from our population survey analysis. This was different from the selection process used to select COVID-19 cases among Diaconat-Fonderie medical records, where patients with positive influenza tests were not retained. Indeed, no case of coinfection by both COVID-19 and influenza was documented in the literature when the retrospective analysis was conducted in the Diaconat-Fonderie laboratory.

Figure 3 shows the rate of negative PCR tests performed by Diaconat-Fonderie laboratory and documented in the population-based survey. Although the headcount was small, the curves show a growing rate of negative tests after 26 January. These data also show that influenza was circulating in the Haut-Rhin population at least until March 2020 and should be considered in the analysis of the survey data.

### 3.2. Comparison of Epidemics among Adults and Children

Existing COVID-19 epidemiological data show that children are significantly less affected than adults [24,25]. Another study conducted in one of the earliest French COVID-19 clusters in the department of Oise has documented that infection in young children (below 12 years old) was largely mild or asymptomatic, but also that other respiratory viruses were circulating concurrently with COVID-19 in the French population in February 2020 [26].

To check whether other respiratory syndromes could explain the pattern observed on Figure 1, we analyzed the number of cases among children aged less than 15 years old in comparison to the rest of the population surveyed. Figure 4 shows the seven-day moving average of daily incidence among adults and children aged 15 years and older (green line) compared to children under 15 years old (orange line) in the surveyed Haut-Rhin households. Although more exposed to respiratory viruses, children below 15 years old displayed significantly fewer symptoms than adults and teenagers over 15 years old. The outbreak existed for children, but in a reduced way, beginning at the second wave and stopping earlier. During the second wave from 23 January to 17 February, the average ratio of the number of cases among adults and children aged 15 years and older to the number of cases among children aged less than 15 was 2.7 (SD = 0.7). During the third wave, from 20 February to 5 April 2020, the same ratio raised to 10 (SD = 3.5). This indicates that other respiratory syndromes were circulating at the end of January among children, but Figure 4 also confirms that the epidemic threshold was crossed 30 January for individuals aged 15 years and older.

### 3.3. Prevalence of Anosmia among COVID-19 Symptoms

A way to discriminate between COVID-19 and other respiratory syndromes is to study symptoms that are more specific to COVID-19. As discussed earlier, one of the most predictive COVID-19 symptoms is anosmia. Out of the 1516 possible cases documented by Haut-Rhin households, 13 out of 158 children less than 11 years old (8.2%), 19 out of 67 children (28.4%) between 12 and 18 years old, and 506 out of 1265 adults (40%) experienced anosmia at different stages.

Figure 5 shows the seven-day moving average of daily incidence for individuals that documented different anosmia intensities from 1 (smell modified) to 3 (total loss of smell). If we considered the total curve (in blue), the 3-wave evolution could be seen, with a crossing of the epidemic threshold on 2 February, a little later than previous estimates. Nevertheless, the speed of the curve was positive since 30 January. The third wave was much more developed and crossed the epidemic threshold on 19 February, with a positive speed since the day before.

The curves corresponding to partial (orange) and total (red) loss of smell display similar patterns, in particular a little increase at the end of January. The curve corresponding to smell modification (yellow) is less marked.

### 3.4. Impact of the POC Gathering

Of particular interest in the case of the Haut-Rhin cluster is the role played by the POC gathering in the outbreak genesis and acceleration in February and beginning of March. The POC gathering began on 17 February and ended 21 February. In the population-based survey answers, 237 out of the 1516 individuals who experienced symptoms commonly attributed to COVID-19 infection come from households in which one member at least participated in the POC gathering.

Figure 6 shows the seven-day moving average of daily incidence for possible COVID-19 cases among Haut-Rhin households with (orange line) or without participants (grey line) to the POC gathering. The two curves display very different shapes: the grey line (households without participants to the POC gathering) shows the 3-wave evolution previously described (see Figure 1), with epidemic thresholds crossed at very similar times. Among the POC gathering attendants, some possible COVID-19 cases were declared at the beginning of February, but there was no clear outbreak. After 21 February, the last day of the gathering, the POC curve is typical of a one-time and one-place outbreak, very close to the pattern observed in the high school cluster in the Oise department in February–March 2020 [3]. The cluster created by the POC gathering participants did not change the speed and acceleration of the non-attendant population during the increasing or decreasing periods. This shows that the two events were concomitant but not strongly linked. Independent of the POC gathering, the outbreak in the Haut-Rhin population began on 26 January and developed by its own. As more than 2000 persons coming from various places of the French territory—including overseas departments and territories—attended the POC meeting, its role in disseminating SARS-CoV-2 to other regions in France will have to be investigated.

Anosmia was observed more frequently (56.5%) in the households involved in the POC meeting than in the other households (32.0%), *p* < 0.0001. Figure 7 shows the seven-day moving average of daily incidence for possible COVID-19 cases with anosmia among Haut-Rhin households that either were involved in the POC gathering (orange line) or were not (grey line). Compared to Figure 1 or Figure 5, it also points to the first epidemic wave end of January.

## 4. Discussion

The results presented in the previous section were obtained from three sources: a population-based survey of the households living the Haut-Rhin department, a retrospective analysis of the Diaconat-Fonderie private, not-for-profit hospital emergency care unit in Mulhouse, and influenza PCR test data from the Diaconat-Fonderie biological laboratory. They all point toward an early circulation of the COVID-19 in the Haut-Rhin population, much earlier than the first official case identified on 26 February 2020 in the Grand Est region and the epidemic alert on 3 March [4,22].

The converging evidence for a COVID-19 epidemic starting at the end of January 2020 comes from the simultaneous epidemic threshold crossings of three independent observables: visitors to Diaconat-Fonderie emergency care service retrospectively diagnosed as probable and confirmed COVID-19 cases (Figure 1 and Figure 2), possible COVID-19 cases from the population survey (Figure 1), and to a lesser extent the increase in negative influenza PCR tests performed by Diaconat-Fonderie (Figure 3).

Our data also provide evidence for the circulation of other respiratory syndromes including influenza, especially during the second wave from 23 January to 17 February 2020. The ratio of the number of retrospectively diagnosed probable and confirmed COVID-19 cases from medical records to the number of possible COVID-19 cases from the population survey was significantly larger in the third wave compared to the second wave (Figure 1). Moreover, the ratio of cases among adults and children was significantly larger in the third wave compared to the second wave (Figure 4). Finally, the occurrence of anosmia among possible COVID-19 cases documented in the survey was also more frequent in the third wave compared to the second wave (Figure 5). However, even if the data collected through the population-based survey include other respiratory syndromes, they trace an early outbreak due to COVID-19. The coexistence of both epidemics, influenza and COVID-19, made the latter more difficult to monitor. According to the French Health ministry influenza observatory [32], seasonal influenza reached its peak nationwide in week 6 (3–9 February 2020). The retrospective analysis of Diaconat-Fonderie medical records and the increase in negative influenza PCR tests starting on 26 January 2020 at the peak of the influenza epidemic suggests that the growth of the COVID-19 epidemic was unnoticed (Figure 2 and Figure 3).

The generation interval computed from intra-household data is close to what is published elsewhere [22,23] and the secondary attack rate is in a range consistent with previously publications [33]. In comparison, the secondary attack rate for influenza is much lower and estimated at 8% [21]. It should be stressed that our estimates of the generation interval and secondary attack rate only reflect the virus dynamics in the context where one infected member continues participating in normal family life.

Finally, our results show very different patterns for households with or without participants to the POC gathering. The curves corresponding to the households related to the POC gathering are very typical of a unique epidemic process, centered on one maximum (Figure 6 and Figure 7). On the same figures, the curves corresponding to households that are not related to the POC gathering show the three wave pattern discussed previously: the first two waves lasted for 24 days and the epidemic threshold was crossed during the second wave.

It is important to note that the Diaconat-Fonderie emergency service is not the main line emergency service in Mulhouse (the main line is at the public hospital). It is more activated when the public hospital is overcrowded: this could be an alternative explanation to the flat epidemic in January and February, before the generalization of the outbreak. The other explanation documented previously is the circulation of other respiratory syndromes in parallel to the start of the COVID-19 epidemic in January and February.

Our data favor the minor role of children in the COVID-19 epidemic. Only limited COVID-19 symptoms were observed among children below 15 years old, especially during the third wave after 20 February.

The population-based survey was also extremely helpful to receive testimonies of people who experienced frequently observed COVID-19 symptoms before the end of 2019. Evidence exists that SARS-CoV-2 was already spreading in France in late December 2019 [10], while the first cases were confirmed in China as early as 8 December 2009 [34]. Alsace is famous internationally for hosting Christmas markets at the end of the year that attract many tourists from all over the world, including tourists from China. This possible scenario should require further investigation.

Our work has some limitations. Possible biases include, for the population-based survey, the exclusion of people without access to internet, dependency on participant declarations, and voluntary participation. Nevertheless, the population of respondents was diverse and included families without symptoms. The studies were retrospective, leading to a possible memorization bias for the population-based survey, and the bias of missing data for the emergency patient cohort.

We had no biological proofs of infection for the population-based survey, and these proofs came very late in the emergency retrospective study, due to the fact that, in France, we were unable to perform PCR tests on a large scale until the end of April 2020. Therefore, respondents to the survey were free to declare symptoms not due to COVID-19. Only 121 (8.0%) of the 1516 possible COVID-19 cases were tested for COVID-19: 34 tests were negative and 87 were positive. The results of serological tests were not available when the datasets were analyzed.

Although Diaconat-Fonderie emergency care services documented probable COVID-19 cases in January, the only biological data used to reach our conclusions were the results of influenza PCR tests. As the hospital laboratory observed a significant increase of negative influenza PCR tests at the end of January 2020, we can hypothesize that the growing number of undiagnosed COVID-19-infected patients resulted in a growing rate of negative influenza tests that went unnoticed.

Additionally, due to privacy rules, information on patient localization was very limited, consisting only in the department in which the household is located, excluding therefore a spatial chaining of cases.

To estimate the epidemic threshold, we could not implement a complex statistical method such as the moving epidemic method (MEM), used to model influenza epidemics based upon historical data from a specific country or region [14]. Indeed, in the case of emerging infectious disease, no historical data are available.

To further document the virus history and propagation, more data are needed, especially from the other emergency services in Mulhouse—particularly the public hospital, which was on the front line during the COVID-19 crisis. Imaging data from radiology units should also help trace the early cases that were unseen during the early days of 2020. Further analysis of the population-based survey should help understand if the early possible COVID-19 cases had common activities or social networks, as households were also invited to document their social contacts.

The role of comorbidities has to be explored. It is also necessary to better understand how the attendants to the POC gathering contributed to the local and nationwide epidemic spread, which lead to a complete French lockdown after 17 March, one month after the beginning of the POC gathering and seven weeks after the outbreak started to become significant. This event is of particular interest to explore the stochasticity and heterogeneity in COVID-19 transmission dynamics. In the hypothesis that COVID-19 transmission is stochastic, dominated by a small number of individuals and driven by super-spreading events, models suggest that the epidemic is not monotonically increasing [35]. Such behavior has already been observed in the case of emerging diseases [36,37] and is suggested by our data on COVID-19.

## 5. Conclusions

Population-based survey and hospital data are consistent and support the hypothesis that the COVID-19 outbreak started as early as the end of January 2020 in the Haut-Rhin department, while other respiratory syndromes including influenza were still circulating in the population.

An efficient sanitary surveillance requires time to collect and analyze data, and to achieve enough convergence to provide an alert. An efficient surveillance system requires about one week [38,39], so a local alert should have taken place around 4 February, but the sanitary alert first occurred on 3 March. This delay was probably due to the fact that surveillance was mainly based on resuscitation ward data, excluding data from general practitioners and data from emergency units, repeating the strategy implemented in 2002–2003 when facing the SARS epidemic. However, during the 2002–2003 SARS epidemic, all patients were severely ill, none was asymptomatic, and there was no transmission of the disease during the incubation period. Surprisingly, the information networks created in France to combine all existing sources in order to monitor the seasonal influenza [32] were not activated for COVID-19. The lack of population-based information was crucial, leading to delayed decisions based on insufficient data. The example of the POC gathering is particularly dramatic: data coming from population-based survey and hospital services confirm that the insufficient sanitary surveillance resulted in a gathering that should have been cancelled two weeks earlier by local sanitary authorities. As about 2000 attendants to this gathering went back to their home, everywhere in France, cancelling this event could have changed the decision of a nationwide lockdown. Nevertheless, the POC gathering did not create the outbreak in the Haut-Rhin department. Its impact on the outbreak evolution requires further investigation. As local events continued being held in a context where the very beginning of the epidemic had remained unseen, the spread was inevitable. Lockdown in Haut-Rhin could have decreased the incidence quickly, but the decision was made too late.

Of course, it could be said that it is easy to re-write history when you know how it turns out. Nevertheless, in order to face a second COVID-19 epidemic wave, our work emphasizes the need for population-based surveys crossed with data from emergency care units, to provide alerts at the best possible time. Delayed decisions always lead to degraded health policies.

## Figures and Tables

**Figure 1 ijerph-17-07175-f001:**
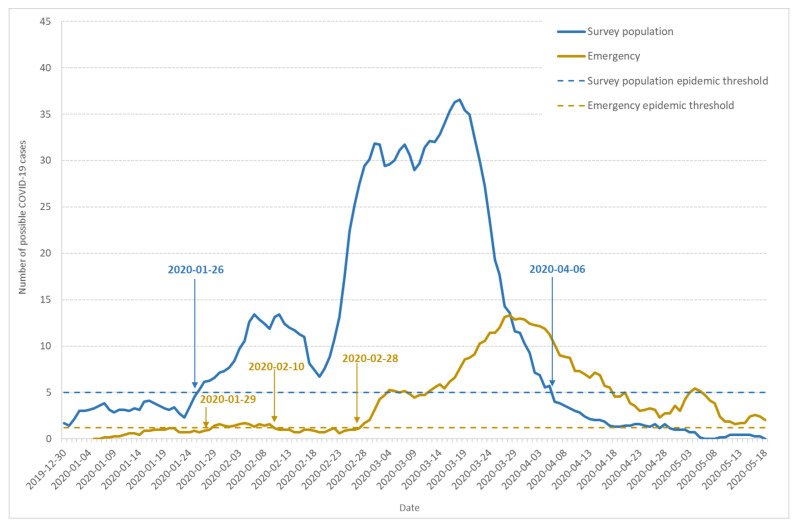
Dynamics of possible COVID-19 cases: seven-day moving average of daily incidence and epidemic threshold. The figure shows data from the population-based survey (blue line), emergency care services (yellow line), and corresponding epidemic thresholds (dashed blue and dashed yellow lines).

**Figure 2 ijerph-17-07175-f002:**
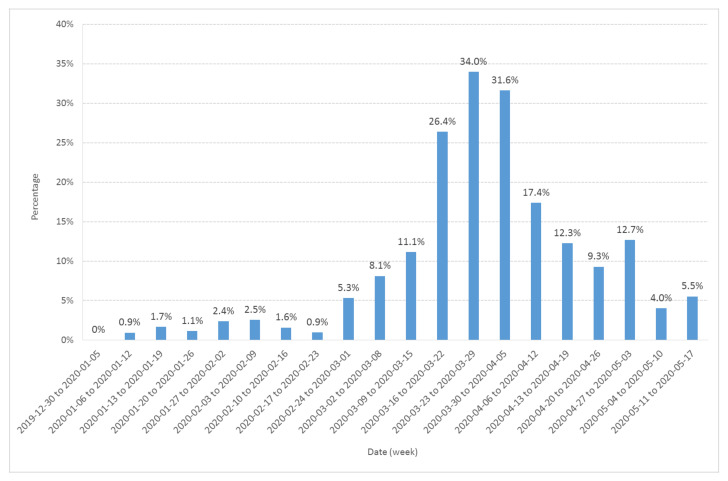
Fraction (expressed in %) of the Diaconat-Fonderie emergency care service weekly visitors retrospectively diagnosed as probable or confirmed COVID-19 cases from 30 December 2019 to 17 May 2020.

**Figure 3 ijerph-17-07175-f003:**
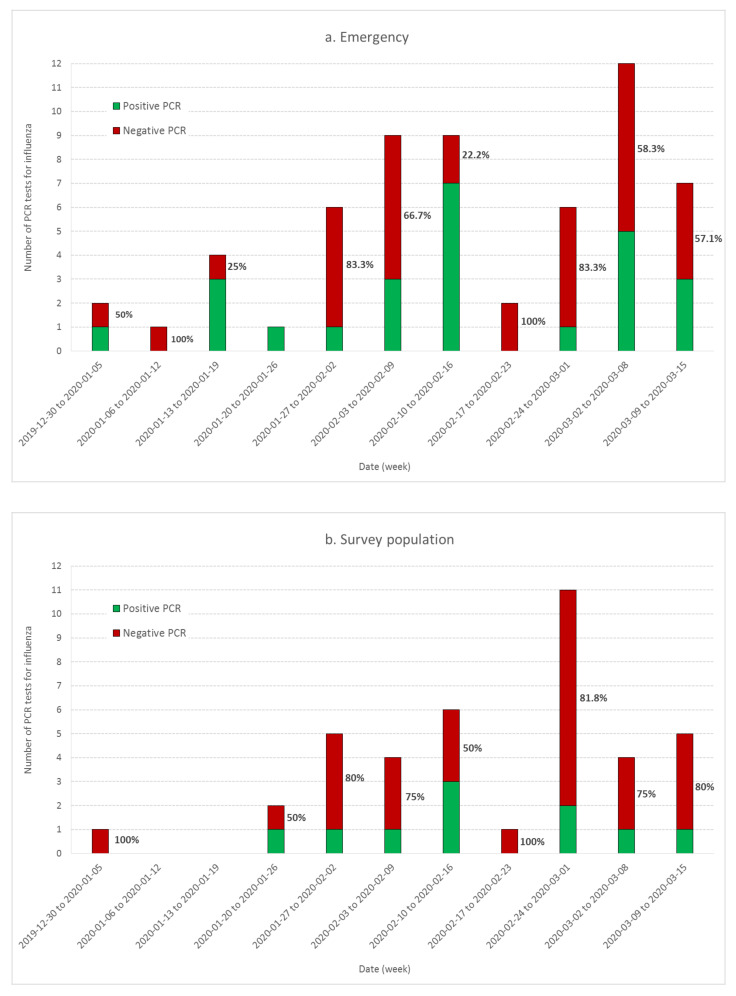
Time evolution of the proportion of influenza PCR positive (green) and negative (red) tests conducted at the Diaconat-Fonderie biology laboratory (**a**) and documented in the population-based survey (**b**). The rate of negative tests as a function of time is documented on each bar.

**Figure 4 ijerph-17-07175-f004:**
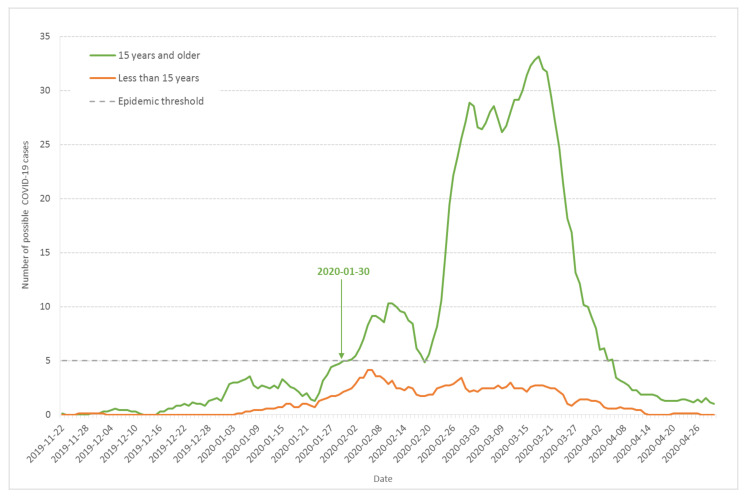
Seven-day moving average of daily incidence of possible COVID-19 cases based on the onset day of the first COVID-19 symptoms for children less than 15 years old (orange line) and adults and children above 15 (green line) from the population-based survey of Haut-Rhin households.

**Figure 5 ijerph-17-07175-f005:**
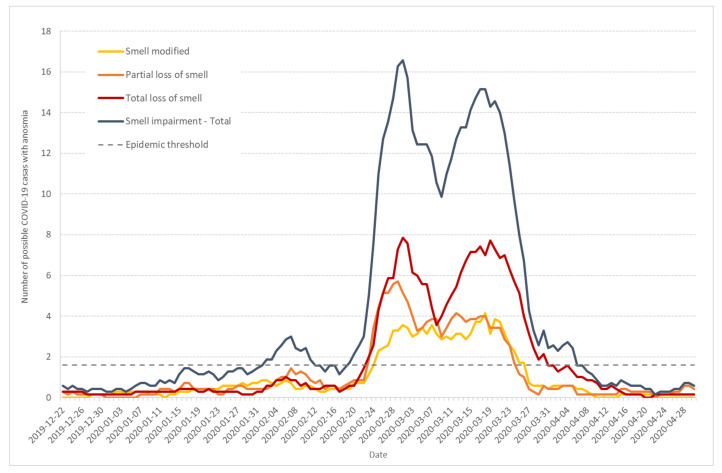
Seven-day moving average of daily incidence of possible COVID-19 cases that documented anosmia with intensities from 1 (modified smell) to 3 (total loss) (from survey data).

**Figure 6 ijerph-17-07175-f006:**
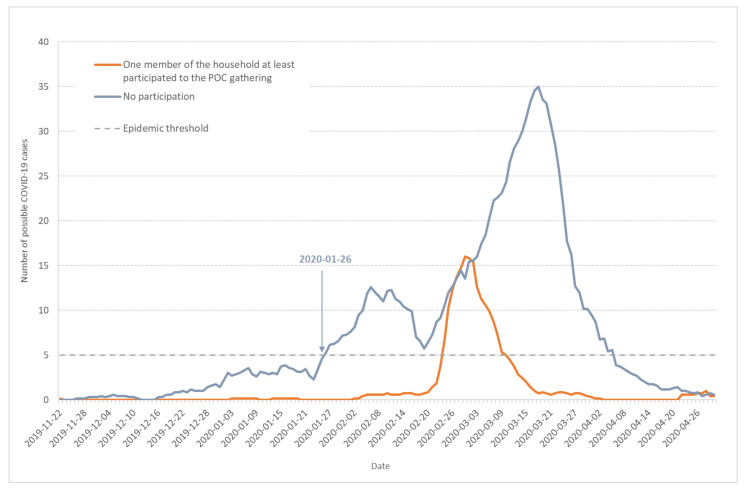
Seven-day moving average of daily incidence for possible COVID-19 cases among Haut-Rhin households. Orange line corresponds to households in which at least one member participated to the POC gathering in Mulhouse (17–21 February 2020) and grey line to households without participants to the POC gathering.

**Figure 7 ijerph-17-07175-f007:**
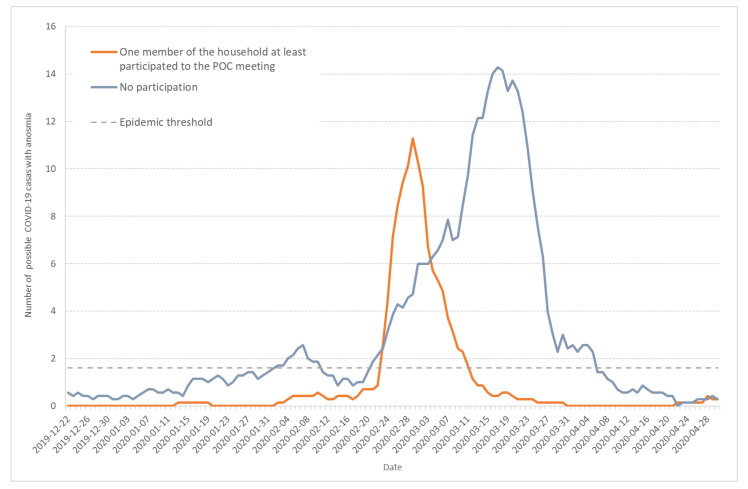
Seven-day moving average of daily incidence for possible COVID-19 cases that documented anosmia among Haut-Rhin households. Orange line corresponds to households in which at least one member participated to the POC gathering in Mulhouse (17–21 February 2020) and grey line to households without participants to the POC gathering.

**Table 1 ijerph-17-07175-t001:** Composition of the 883 households in the Haut-Rhin department with at least one possible COVID-19 case.

Household Composition	*n*
Household size, mean (SD)	2.8 (1.2)
Number of adults in the household, mean (SD)	2.2 (0.9)
Age of adults (years), mean (SD)	44.4 (16.9)
Children in the household, *n* (%)	
No	552 (62.5)
Yes	331 (37.5)
Number of children in the household, mean (SD)	1.7 (0.7)
Age of children (years), mean (SD)	7.5 (4.1)
Number of possible cases of COVID-19 infection in the household, mean (SD)	1.7 (0.9)

Adults are 15 years and older; children are less than 15 years old.

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
