# Peer review of "Hospital and Population-Based Evidence for COVID-19 Early Circulation in the East of France"

_ijerph, 2020, doi:10.3390/ijerph17197175_

Round 1

Reviewer 1 Report

This is an interesting analysis of survey data, emergency department data, and influenza testing data that aims to estimate the early arrival of SARS-CoV-2 infection in Eastern France. There are a number of things that could be done to improve the paper

  1. Though the writing is understandable, it needs significant editing for grammar and usage. In particular, terms such as the seven day moving average are referred to several different ways throughout the paper, but in my experience should be written as above. 
  2. The reliance on the number of negative influenza tests doesn't make sense to me, since we would expect the number of negative tests to go up even if the illness is due to influenza. More influenza would lead to more testing, including more negative tests, since the rate of positive tests is still relatively low. At the peak of the flu season in the US, the positivity rate was 30%. What should be measured is the rate of negative tests, which we'd expect to go up if lots of people show up with influenza symptoms and test negative. 
  3. I think several of the conclusions could have alternative explanations and those should be acknowledged. For instance, there is a statement that the POC cluster didn't change the speed and acceleration of the non-attendant population, but when I look at that graph it seems plausible that the POC cluster could have contributed to the continued rise in other cases. In addition, the first rise in cases in late January seems related in time to the rise in positive influenza tests (Figure 3) and I'd interpret that first rise as just as likely due to influenza. Anosmia can also be reported with influenza, granted at a lower rate than COVID-19. 
  4. On line 249 you report that you didn't exclude respondents that tested positive for influenza from the survey analysis, but on line 110 you say that you did exclude them from the emergency care analysis. Shouldn't they be treated the same in both data sets?
  5. When comparing the number of individuals with COVID symptoms by age group (Figure 4) it would be more informative to use rates instead of numbers. Related to comment #3, I would interpret the rise in cases < 15 years starting in late January as evidence supporting that this first 'wave' was due to influenza. 
  6. The discussion of transmission dynamics in lines 365-371, and the statement that these data confirm the minor role of children in the epidemic are not supported by the information in the paper. In general the conclusions go beyond what is documented. 
  7. Some discussion of the earliest documented cases in China should be included to give some context to the likelihood of early spread in France. 

Author Response

We highly appreciate the relevance of the comments made by the reviewers and have prepared a revised version of the manuscript that will hopefully improve its quality and make it suitable for publication in this journal. The editorial changes are written in green in order to ease the reviewer work.

Please find below the detail of our responses to the comments expressed by reviewer n°1.

First of all, we are thankful for the interest expressed for the paper.  

Point 1: Though the writing is understandable, it needs significant editing for grammar and usage. In particular, terms such as the seven day moving average are referred to several different ways throughout the paper, but in my experience should be written as above. 

Response 1: we have done a complete editing of the manuscript before resubmission.

Change to the manuscript to address point 1

Many changes have been made to improve grammar and usage. They are all documented using a green police.

Point 2: The reliance on the number of negative influenza tests doesn't make sense to me, since we would expect the number of negative tests to go up even if the illness is due to influenza. More influenza would lead to more testing, including more negative tests, since the rate of positive tests is still relatively low. At the peak of the flu season in the US, the positivity rate was 30%. What should be measured is the rate of negative tests, which we'd expect to go up if lots of people show up with influenza symptoms and test negative. 

Response 2: we thank the reviewer for pointing the lack of clarity in our explanations. The influenza tests discussed in the article were performed by the hospital laboratory upon request of emergency care physicians for patients displaying flu-like clinical symptoms in order to confirm their diagnosis. Only a subset of the physicians has adopted this diagnosis strategy because performing PCR tests is time consuming in a context of emergency care. Two third (66%) of the influenza tests performed before January 29th were positive. The rate of negative tests started to rise from January 29th till mid-March 2020 when emergency care physicians stopped requesting flu PCR tests and only asked for COVID-19 PCR tests.

Change to the manuscript to address point 2

We edited the introduction to provide more context about the flu tests performed by the Diaconat-Fonderie laboratory.

In section 3.1.3, we have changed Figure 3 and edited the paragraph. Discussion in section 4 has also been modified to introduce more caution in the results interpretation.

Point 3: I think several of the conclusions could have alternative explanations and those should be acknowledged. For instance, there is a statement that the POC cluster didn't change the speed and acceleration of the non-attendant population, but when I look at that graph it seems plausible that the POC cluster could have contributed to the continued rise in other cases. In addition, the first rise in cases in late January seems related in time to the rise in positive influenza tests (Figure 3) and I'd interpret that first rise as just as likely due to influenza. Anosmia can also be reported with influenza, granted at a lower rate than COVID-19.

Response 3:  we agree with the reviewer that some of the conclusions were not fully supported by the data.

Regarding the observed rise in the number of cases end of January, it should be kept in mind that the retrospective analysis of Diaconat Fonderie medical records also document the rise observed in survey data although records with positive influenza PCR tests were not retained as COVID-19 potential cases.

Regarding anosmia, several articles provided evidences that this symptom was frequently observed in case of SARS-CoV2 infection and could be used as a relevant screening tool to help identify people with potential mild cases (see for instance Menni, C., Valdes, A.M., Freidin, M.B. et al. Real-time tracking of self-reported symptoms to predict potential COVID-19. Nat Med 26, 1037–1040 (2020). https://doi.org/10.1038/s41591-020-0916-2). However, it was not used by the emergency care service in their retrospective analysis because it was not known in the period of interest to our study (January to April 2020).

Change to the manuscript to address point 3

To address this point, we have edited the results interpretation (section 3), the discussion (section 4) and the conclusion (section 5).

Point 4: on line 249 you report that you didn't exclude respondents that tested positive for influenza from the survey analysis, but on line 110 you say that you did exclude them from the emergency care analysis. Shouldn't they be treated the same in both data sets?

Response 4: we thank the reviewer for pointing this difference. At the time the Diaconat-Fonderie emergency service did its retrospective analysis, no case of coinfection by both COVID-19 and flu had been documented in the literature. But such cases have been observed (see for instance Cuadrado-Payán E, Torres-Elorza EMontagud-Marrahi M, Bodro M, Blasco M, Poch E, Soriano A, Gas-ton  J,  Piñeiro  (2020)  SARS-CoV-2  and  influenza  virus  co-infection.  Lancet  395:e84.  https ://doi.org/10.1016/S0140 -6736(20)31052 -7 ) and we therefore chose to keep them in our survey analysis. It should be stressed that only 12 persons were tested positive for influenza.

Change to the manuscript to address point 4

We added explanations and the reference above to clarify this confusing point.

Point 5: When comparing the number of individuals with COVID symptoms by age group (Figure 4) it would be more informative to use rates instead of numbers. Related to comment #3, I would interpret the rise in cases < 15 years starting in late January as evidence supporting that this first 'wave' was due to influenza. 

Response 5: comparing evidences from the three sources, we agree that the rise late January mixes COVID-19 and other respiratory syndromes including influenza.

The evidence for the circulation of other respiratory syndromes is the increase in the number of cases among children less than 15 years. The main evidences for COVID-19 circulation during the same period are the increase in the number of COVID-19 cases from medical records and the increase in anosmia cases among survey answers.

Change to the manuscript to address point 5

To address this point, we have edited the results interpretation (section 3), the discussion (section 4) and the conclusion (section 5).

We have computed and documented ratios of cases for the different age groups.

Point 6: The discussion of transmission dynamics in lines 365-371, and the statement that these data confirm the minor role of children in the epidemic are not supported by the information in the paper. In general the conclusions go beyond what is documented. 

Response 6: we have taken this point into account by revising the conclusions and refocussing it on the documented evidences.

Change to the manuscript to address point 6

A number of changes have been introduced in the discussion and conclusion sections to consider the reviewer remarks.

Point 7: Some discussion of the earliest documented cases in China should be included to give some context to the likelihood of early spread in France. 

Response 7: we added to the references an article about the early transmission dynamics of the virus in China (Q. Li, X. Guan, P. Wu, X. Wang, L. Zhou, Y. Tong, R. Ren, K.S.M. Leung, E.H.Y. Lau, J.Y. Wony, X. Xing, N. Xiang, Y. Wu, C. Li, Q. Chen, D. Li, T. Liu, J. Zhao, M. Liu, W. Tu, C. Chen, L. Jin, R. Yang, Q. Wang, S. Zhou, R. Wang, H. Liu, Y. Luo, Y. Liu, G. Shao, H. Li, Z. Tao, Y. Yang, Z. Deng, B. Liu, Z. Ma, Y. Zhang, G. Shi, T.T.Y. Lam, J.T. Wu, G.F. Gao, B.J. Cowling, B. Yang, G.M. Leung, Z. Feng, Early transmission dynamics in Wuhan, China, of novel coronavirus-infected pneumonia, N. Engl. J. Med., 382 (2020), pp. 1199-1207). This article documents the first 425 confirmed cases of COVID-19 infections in Wuhan. The majority of the earliest cases reported exposure to the Huanan Seafood Wholesale Market, but nonlinked cases are documented as early as beginning of December 2019. The mean duration from onset to hospital admission was estimated to be 12.5 days (95% CI, 10.3 to 14.8) among the 44 cases with illness onset before January 2020.

Our present knowledge of the virus transmission dynamics and proportion of asymptomatic individuals opens the possibility that asymptomatic or pauci-symptomatic cases could travel to France, and especially to Alsace, as early as December 2019 or January 2020. A special opportunity for such international travels are Alsace Christmas markets which attract many Chinese tourists before Christmas. Further retrospective studies will be needed to explore this possible scenario.

Change to the manuscript to address point 7

We added a reference to the references (Q. Li et al, Early transmission dynamics in Wuhan, China, of novel coronavirus-infected pneumonia, N. Engl. J. Med., 382 (2020), pp. 1199-1207) and a paragraph in the discussion section.

Reviewer 2 Report

    The manuscript " Hospital and population-based evidences for COVID-19 early circulation in the East of France" use the integration of data coming from multiple sources to help to understand the SARS-CoV-2 dynamics and transmission. It has certain indicative significance in help an early alarm in an emerging disease. However, there are a couple of things I feel need to be improved.

    1.What other similar questionnaire-based studies do? The author should cite some more references.

    2.Figure 1-7 horizontal and vertical coordinates, lack names, the author should explain the specific meaning and make modifications.

    3.There are some writing errors:

    Lines 195 “..”

    Lines 305 and Lines 330 “grey lines correspond to households without participants to the POC gathering in Mulhouse?”

Author Response

We highly appreciate the relevance of the comments made by the reviewers and have prepared a revised version of the manuscript that will hopefully improve its quality and make it suitable for publication in this journal.

Please find below the detail of our responses to the comments expressed by reviewer n°2.

First of all, we are thankful for the interest expressed for the paper.  

Point 1.What other similar questionnaire-based studies do? The author should cite some more references.

Response 1: we thank the reviewer for raising this point. Population-based surveys are frequently used in social sciences but also in epidemiology with a particular emphasis on respiratory syndroms (see for instance: Social contact patterns relevant to the spread of respiratory infectious diseases in Hong Kong. Sci Rep 7, 7974 (2017) https://doi.org/10.1038/s41598-017-08241-1)

Change to the manuscript to address point 1

We have added in the introduction a paragraph and three references about the relevance of population-based surveys to study the epidemiology of respiratory syndromes.

Point 2. Figure 1-7 horizontal and vertical coordinates, lack names, the author should explain the specific meaning and make modifications.

Response 2: we have modified the figures to address the reviewer remark.

Change to the manuscript to address point 2: axis labels have been added on figures 1 to 7.

Point 3. There are some writing errors:

    Lines 195 “..”

    Lines 305 and Lines 330 “grey lines correspond to households without participants to the POC gathering in Mulhouse?”

Response 3: we thank the reviewer for pointing these typos.

Change to the manuscript to address point 3: typos have been corrected.

Reviewer 3 Report

This paper performed retrospective observational data review for regional COVID-19 outbreak in France and indicated similar insights on the epidemic shared through different data sources. Most of their findings resonate with the knowledge from ongoing investigation of transmission, epidemiology and clinical aspects of SARS-CoV-2, and stressed the importance of surveillance monitoring of emerging outbreak events. The combination of online survey data, medical records and social behavior logging presented in the paper again indicated infectious disease outbreaks could have huge impact on many aspects of our daily life, and we could trace back the transmission events from numerous source of information.

I have a few major comments for potential improvement of the manuscript:

  1. The conclusion of commonality among epidemic curves obtained from the 3 data sources do exist as shown in the figures. There was much stronger correlation between household survey data and emergency service data, but influenza PCR test results were much less associated with the other two. It's likely that influenza tests be related to COVID disease as mentioned in the paper, however the current result did not provide significant link.
  2. It is not clear how the author derived the "epidemic threshold" line 154-155, and why it could be used for indication of ongoing outbreak. I'd like to see more discussion on the rationale behind this and potentially references?
  3. Line 218-219, more details needed on how the calculation is performed to calculate R and generation interval using the household data. As discussed line 351-360, the analysis approach for household clustered data vs. general population outbreak data is quite different. It's much easier to estimate the secondary attack rate with household data, especially e.g. between adults or adult-children transmission rates, than estimating the general R in community with household data only. The contact structure/behavior is quite specific within households.
  4. The presentation on anosmia prevalence is speculative. I think this part is not informative enough on the outbreak trend itself, as shown in fig. 5 the proportional trend for different severity remains almost constant. It might be clinically important with a bit higher specificity, however not serving well on the major findings of current study.
  5. Line 361-371, POC gathering is an important transmission event in the area. However this did not support the conclusion that "transmission is driven by super-spreading events" without further data analysis.
  6. Line 381-387, I'm confused how this contributes to the paper?

And some minor comments:

  1. Fig.1, the legends "population" and "emergency", which actually means data "by survey" and "by medical records". "Population" could be confused with epidemic trend published by local government as official number.
  2. Fig.2, assuming the overall attendance number remain stable, the COVID-19 related attendance proportion trend would be highly correlated with survey data, is this true? Otherwise the absolute number may show a different trend.
  3. Fig.3, could these trend be compared with that from a different year? Then it may indicate it's COVID related.

Author Response

We highly appreciate the relevance of the comments made by the reviewers and have prepared a revised version of the manuscript that will hopefully improve its quality and make it suitable for publication in this journal. The editorial changes are written in green in order to ease the reviewer work.

Please find below the detail of our responses to the comments expressed by reviewer n°3.

First of all, we are thankful for the interest expressed for the paper.  

Point 1: The conclusion of commonality among epidemic curves obtained from the 3 data sources do exist as shown in the figures. There was much stronger correlation between household survey data and emergency service data, but influenza PCR test results were much less associated with the other two. It's likely that influenza tests be related to COVID disease as mentioned in the paper, however the current result did not provide significant link.

Response 1: we thank the reviewer for raising this point. We agree that influenza PCR test results are less significant than the results coming from the other data sources (survey results and medical records). The influenza tests discussed in the article were performed by the hospital laboratory upon request of emergency care physicians for patients displaying flu-like clinical symptoms in order to confirm their diagnosis. Only a subset of the physicians has adopted this diagnosis strategy because performing PCR tests is time consuming in a context of emergency care. Two third (66%) of the influenza tests performed before January 29th were positive. The rate of negative tests started to rise from January 29th till mid-March 2020 when emergency care physicians stopped requesting flu PCR tests and only asked for COVID-19 PCR tests.

Change to the manuscript to address point 1

We edited the introduction to provide more context about the influenza PCR tests performed by the Diaconat-Fonderie laboratory.

In section 3.1.3, we have changed Figure 3 and edited the paragraph. Discussion in section 4 has also been edited to introduce more caution in the results interpretation.

Point 2: It is not clear how the author derived the "epidemic threshold" line 154-155, and why it could be used for indication of ongoing outbreak. I'd like to see more discussion on the rationale behind this and potentially references?

Response 2: we thank the reviewer for raising this point. Very similar to the method used in (Costagliola, D et al. “A routine tool for detection and assessment of epidemics of influenza-like syndromes in France.” American journal of public health vol. 81,1 (1991): 97-9. doi:10.2105/ajph.81.1.97) for influenza-like syndromes, we first computed a non-epidemic level. For survey data, the non-epidemic level was defined as the upper limit of the seven day moving average of daily incidence over the period December 15th 2019 to January 15th 2020. For Diaconat-Fonderie medical records, it was defined as the upper limit of the seven day moving average of daily COVID-19 cases over the period January 1st to January 15th 2020.

For both data sets, the threshold was then defined as the upper 95% confidence limit of this non-epidemic level. We checked for both datasets the stability in terms of consecutive days over or under the threshold. 

Change to the manuscript to address point 2

We have added a paragraph and 2 references in section 2.2.

Point 3: Line 218-219, more details needed on how the calculation is performed to calculate R and generation interval using the household data. As discussed line 351-360, the analysis approach for household clustered data vs. general population outbreak data is quite different. It's much easier to estimate the secondary attack rate with household data, especially e.g. between adults or adult-children transmission rates, than estimating the general R in community with household data only. The contact structure/behavior is quite specific within households.

Response 3: we thank the reviewer for raising this point. Indeed, computing the secondary attack rate is much more relevant as our survey data provide information on transmission in households. As a consequence, we have computed and discussed the secondary attack rate instead of the reproduction number in the revised manuscript.  

Change to the manuscript to address point 3

Definition and references to the secondary attack rate have been added in section 2. A paragraph has been added to discuss our results in section 4.

Point 4: The presentation on anosmia prevalence is speculative. I think this part is not informative enough on the outbreak trend itself, as shown in fig. 5 the proportional trend for different severity remains almost constant. It might be clinically important with a bit higher specificity, however not serving well on the major findings of current study.

Response 4: we thank the reviewer for raising this point. Several articles provided evidences that this symptom was frequently observed in case of SARS-CoV2 infection and could be used as a relevant screening tool to help identify people with potential mild cases (see for instance Menni, C., Valdes, A.M., Freidin, M.B. et al. Real-time tracking of self-reported symptoms to predict potential COVID-19. Nat Med 26, 1037–1040 (2020). https://doi.org/10.1038/s41591-020-0916-2). However, it was not used by the emergency care service in their retrospective analysis because it was not known in the period of interest to our study (January to April 2020).

Change to the manuscript to address point 4

We have added the reference above to further stress the relevance of anosmia to discriminate COVID-19 from other respiratory syndromes.

We have also edited section 3.3 and further discussed anosmia results in section 4.

Point 5: Line 361-371, POC gathering is an important transmission event in the area. However this did not support the conclusion that "transmission is driven by super-spreading events" without further data analysis.

Response 5: we agree with the reviewer that this statement is preliminary before further data analysis.

Change to the manuscript to address point 5

We edited the discussion section and only mention the interest of further studying the POC gathering as a potential Super Spreading Event.

Point 6: Line 381-387, I'm confused how this contributes to the paper?

Response 6: the goal was to explain how the survey was also an opportunity to access information that was not foreseen initially, especially about the virus early circulation in Alsace.

Change to the manuscript to address point 6

We took away part of this paragraph, keeping only a discussion on the likelihood of early spread in France. We added to the references an article about the early transmission dynamics of the virus in China (Q. Li, X. Guan, P. Wu, X. Wang, L. Zhou, Y. Tong, R. Ren, K.S.M. Leung, E.H.Y. Lau, J.Y. Wony, X. Xing, N. Xiang, Y. Wu, C. Li, Q. Chen, D. Li, T. Liu, J. Zhao, M. Liu, W. Tu, C. Chen, L. Jin, R. Yang, Q. Wang, S. Zhou, R. Wang, H. Liu, Y. Luo, Y. Liu, G. Shao, H. Li, Z. Tao, Y. Yang, Z. Deng, B. Liu, Z. Ma, Y. Zhang, G. Shi, T.T.Y. Lam, J.T. Wu, G.F. Gao, B.J. Cowling, B. Yang, G.M. Leung, Z. Feng, Early transmission dynamics in Wuhan, China, of novel coronavirus-infected pneumonia, N. Engl. J. Med., 382 (2020), pp. 1199-1207).

Our present knowledge of the virus transmission dynamics and proportion of asymptomatic individuals opens the possibility that asymptomatic or pauci-symptomatic cases could travel to France, and especially to Alsace, as early as December 2019 or January 2020. A special opportunity for such international travels are Alsace Christmas markets which attract many Chinese tourists before Christmas. Further retrospective studies are of course needed to explore this possible scenario.

Point 7: Fig.1, the legends "population" and "emergency", which actually means data "by survey" and "by medical records". "Population" could be confused with epidemic trend published by local government as official number.

Response 7: we thank the reviewer for raising this confusing point. We have replaced “Population” by “Survey population” in all figure legends.

Change to the manuscript to address point 7: All figure legends have been modified as suggested.

Point 8: Fig.2, assuming the overall attendance number remain stable, the COVID-19 related attendance proportion trend would be highly correlated with survey data, is this true? Otherwise the absolute number may show a different trend.

Response 8: yes indeed, if the attendance number remains stable, the COVID-19 related attendance proportion trend is expected to be highly correlated with survey data.
Please find below the absolute number of COVID-19 cases, showing a pattern similar to the one observed in survey data. The first wave end of January is less pronounced, which favors circulation of other respiratory syndromes during that period.

In addition, it is interesting to note that the absolute number of visitors to the emergency unit decreased significantly (a factor 2) during confinement as people were scared of COVID-19.

Change to the manuscript to address point 8

We did not apply changes to the manuscript but are available to do so if recommended by the reviewer.

Point 9: Fig.3, could these trends be compared with that from a different year? Then it may indicate it's COVID related.

Response 9: we agree with the interest of such comparison. However, it is not possible to document trends from previous years because the emergency care service only started this year to ask for influenza PCR tests to support diagnosis in specific clinical contexts…  

Change to the manuscript to address point 9

To improve its significance, Figure 3 was modified to show more explicitly the rate of negative tests as a function of time.

Round 2

Reviewer 1 Report

I appreciate the attention to previous comments. There are several suggestions for additional improvements:

1) Was the abstract modified to be consistent with the modifications to the paper? For instance, in the abstract it is stated as fact that SARS-CoV-2 was circulating several weeks before 2/26, when the data presented 'suggest' this may have been the case, but don't prove it.

2) Are 'confirmed' COVID-19 cases from the emergency department defined in the methods? I see probable cases defined there (line 116) but a clear delineation between probable and confirmed is missing. Typically confirmed cases only get this designation with a laboratory test. These should be referred to as 'retrospectively diagnosed probable cases' or some other similar designation in the text and in Figure 2. Referring to them as diagnosed cases is misleading. 

3) Figure 3 doesn't show a rate, but the text refers to a rate. Please add a negative test rate to the graph above each bar, or group of bars.

4) Line 387 uses the term 'confirms' but I think this is suggestive evidence so 'suggests' would be a better word.

5) Line 430 refers to the number of negative tests for influenza, when its the rate of negative tests that is the more important measure. 

Author Response

We thank the reviewer for his positive evaluation of the changes made to the manuscript following the first review.

The additional editorial changes are written in red in order to ease the reviewer work.

Please find below the detail of our responses to the comments expressed by the reviewer.

Specific points

  • Point 1: Was the abstract modified to be consistent with the modifications to the paper? For instance, in the abstract it is stated as fact that SARS-CoV-2 was circulating several weeks before 2/26, when the data presented 'suggest' this may have been the case, but don't prove it.

Response 1: indeed, we had not modified the abstract. We have revised the abstract in order to be consistent with the changes made in the paper following the first round of revision.  

Change to the manuscript to address point 1: The paragraph on results was modified as recommended (line 23 and lines 25-26). We used “our results suggest” as recommended and “seems to have played” instead of “played”.

  • Point 2: Are 'confirmed' COVID-19 cases from the emergency department defined in the methods? I see probable cases defined there (line 116) but a clear delineation between probable and confirmed is missing. Typically confirmed cases only get this designation with a laboratory test. These should be referred to as 'retrospectively diagnosed probable cases' or some other similar designation in the text and in Figure 2. Referring to them as diagnosed cases is misleading. 

Response 2: We thank the reviewer for raising this important point. Indeed, we defined probable cases of COVID-19, but confirmed cases were not clearly defined. Confirmed COVID-19 patients were patients with chest images indicating typical CT features of COVID-19 pneumonia and/or a positive biological test for COVID-19. Patients were classified as confirmed cases only after March 3rd 2020 when biological tests were available.

Change to the manuscript to address point 2: We added a clearer definition of confirmed cases (lines 136 to 138). We also changed our designation to 'retrospectively diagnosed probable cases' in several occurrences (lines 139, 246, 256, 261, 377, 383) and on figure 2 legend (line 262). 

  • Point 3: Figure 3 doesn't show a rate, but the text refers to a rate. Please add a negative test rate to the graph above each bar, or group of bars.

Response 3: We have modified Figure 3 to address the reviewer remark.

Change to the manuscript to address point 3: On figure 3, the negative PCR test rate was added on each bar. Figure 3 legend was modified accordingly (line 287).

  • Point 4: Line 387 uses the term 'confirms' but I think this is suggestive evidence so 'suggests' would be a better word.

Response 4: We thank the reviewer for raising this point. We agree with his remark.

Change to the manuscript to address point 4: The terms “confirms” was replaced by “suggests” (line 395).

  • Point 5: Line 430 refers to the number of negative tests for influenza, when it is the rate of negative tests that is the more important measure. 

Response 5: We thank the reviewer for pointing this point. We also agree with this comment.

Change to the manuscript to address point 5: The term “number” was replaced by “rate” (line 438).

English editing

To respond to the reviewer requirement for extensive editing of English language and style, the newly revised version of the manuscript has been submitted to MDPI English Editing Service, to improve grammar and phrasing.